# Ecological, Genetic, and Phylogenetic Aspects of YFV 2017–2019 Spread in Rio de Janeiro State

**DOI:** 10.3390/v15020437

**Published:** 2023-02-04

**Authors:** Ieda Pereira Ribeiro, Edson Delatorre, Filipe Vieira Santos de Abreu, Alexandre Araújo Cunha dos Santos, Nathália Dias Furtado, Anielly Ferreira-de-Brito, Anielle de Pina-Costa, Maycon Sebastião Alberto Santos Neves, Márcia Gonçalves de Castro, Monique de Albuquerque Motta, Patricia Brasil, Ricardo Lourenço-de-Oliveira, Myrna Cristina Bonaldo

**Affiliations:** 1Laboratório de Biologia Molecular de Flavivírus, Instituto Oswaldo Cruz, Fundação Oswaldo Cruz, Rio de Janeiro 21040-900, RJ, Brazil; 2Laboratório de Genômica Evolutiva e Ambiental, Departamento de Biologia, Centro de Ciências Exatas, Naturais e da Saúde, Universidade Federal do Espírito Santo, Alegre 29500-000, ES, Brazil; 3Laboratório de Mosquitos Transmissores de Hematozoários, Instituto Oswaldo Cruz, Fundação Oswaldo Cruz, Rio de Janeiro 21040-900, RJ, Brazil; 4Instituto Federal do Norte de Minas Gerais, Salinas 39560-000, MG, Brazil; 5Laboratório de Doenças Febris Agudas, Instituto Nacional de Infectologia Evandro Chagas, Fundação Oswaldo Cruz, Rio de Janeiro 21040-900, RJ, Brazil; 6Faculdade de Medicina de Teresópolis, Centro Universitário Serra dos Órgãos, UNIFESO, Teresópolis 25955-001, RJ, Brazil

**Keywords:** yellow fever virus, Brazilian outbreak, epidemiology, ecology, amino acid polymorphisms, genomic surveillance

## Abstract

In Brazil, a yellow fever (YF) outbreak was reported in areas considered YF-free for decades. The low vaccination coverage and the increasing forest fragmentation, with the wide distribution of vector mosquitoes, have been related to yellow fever virus (YFV) transmission beyond endemic areas since 2016. Aiming to elucidate the molecular and phylogenetic aspects of YFV spread on a local scale, we generated 43 new YFV genomes sampled from humans, non-human primates (NHP), and primarily, mosquitoes from highly heterogenic areas in 15 localities from Rio de Janeiro (RJ) state during the YFV 2016–2019 outbreak in southeast Brazil. Our analysis revealed that the genetic diversity and spatial distribution of the sylvatic transmission of YFV in RJ originated from at least two introductions and followed two chains of dissemination, here named the YFV RJ-I and YFV RJ-II clades. They moved with similar dispersal speeds from the north to the south of the RJ state in parallel directions, separated by the Serra do Mar Mountain chain, with YFV RJ-I invading the north coast of São Paulo state. The YFV RJ-I clade showed a more significant heterogeneity across the entire polyprotein. The YFV RJ-II clade, with only two amino acid polymorphisms, mapped at NS1 (I1086V), present only in mosquitoes at the same locality and NS4A (I2176V), shared by all YFV clade RJ-II, suggests a recent clustering of YFV isolates collected from different hosts. Our analyses strengthen the role of surveillance, genomic analyses of YVF isolated from other hosts, and environmental studies into the strategies to forecast, control, and prevent yellow fever outbreaks.

## 1. Introduction

Yellow fever (YF) continues to be a significant health problem in Brazil. During the second half of the 20th century, its etiological agent, the yellow fever virus (YFV), has been primarily maintained in the country by a sylvatic transmission cycle between non-human primates (NHPs) and forest-canopy mosquitoes from the endemic northern Amazon region [1,2]. However, since the early 2000s, many NHPs and human YFV infections have been reported in the inland portions of the southeastern and southern regions [3]. Curiously, the easternmost southeast zone was still a YFV-free zone for decades [4].

The recent spread of YFV in the entire southeast and central and eastern part of the south region from 2016 to 2019 has caused thousands of deaths of humans and NHP living close to the most densely populated areas of the Atlantic coast, originating the most severe Brazilian outbreak of the last 80 years [5,6,7,8,9,10,11]. Between December 2016 and June 2019, the Brazilian Ministry of Health confirmed 2237 human cases of YF, with 759 deaths and 1567 epizootic events in NHPs across Brazil [12,13,14]. The low vaccine coverage that is due to no recommendations for vaccination in several regions of the country, besides several environmental conditions and ecological factors, was involved in that outbreak [2,11]. The high density of susceptible NHPs, particularly of the genera *Sapajus*, *Alouatta*, and *Callithrix*, found in areas of the Atlantic Forest, combined with the large distribution, high abundance, and natural infection of the primary vectors, *Haemagogus janthinomys*/*capricornii* and *Hg. leucocelaenus,* was responsible for the re-emergence of sylvatic yellow fever (SYF) in the southeast of Brazil [15,16]. Intense YFV circulation generating mutational changes and promoting the emergence of new virus strains has been described [5,17,18,19,20]. The impact of these new lineages on viral fitness in hosts, vectors, and, mainly, vaccine efficacy is unpredictable and has been investigated [21,22,23,24]. Herewith, avoiding the urban YF re-emergence is imperative.

Since the mid-1940s, no human cases of YF have been reported in RJ. Furthermore, there was no evidence of the enzootic circulation of YFV until the arrival of the epidemic wave during the summer of 2016–2017 in the northern state portion [2,19,25]. Notwithstanding the intense vaccination campaign promptly implemented in early 2017, RJ recorded 105 human deaths during two successive transmission seasons that affected the north-center (season 2016–2017) and the center-south (season 2017–2018) [1,12,13,14,26]. Few signs of viral circulation were detected during the third transmission season (2018–2019) in municipalities of RJ that had been touched in the previous two seasons, with no evidence of transmission so far [1,27]. The state encompasses one of the largest metropolitan areas in South America, with 17.2 million inhabitants and numerous domestic and international tourist destinations that turn it into a territory with a great movement of people and goods [26]. However, multiple large less-populous counties of RJ are interspersed with patches of the preserved biodiversity-rich rain forest of variable extents. The state exhibits a peculiar ecological mosaic [28,29,30]. In RJ territory, forest areas with different distances from each other are associated with a diversity of altitudes and seasonal temperature differences. Additionally, the long mountain chain (Serra do Mar) longitudinally divides the state of 43.7 thousand km^2^ into two main primary river basins and impacts rainfall [29,30]. For this reason, analyzing a higher number of virus sequences circulated in such a particular and variable ecological region is of great interest. It constitutes a unique opportunity to enlighten the molecular, genetic, and phylogenetic aspects of YFV spread in virgin zones on a more local scale.

The definition of several YFV 2016–2019 genomes during the southeastern transmission wave has allowed us to understand better the routes of viral introduction in different areas, the speed rate of YFV geographical dispersion, and the molecular epidemiology of YFV during the largest outbreak of YF of the 21st century in the Americas [6,8,9,17,18,19,20,31,32]. Our previous phylogeographic study supports that the most recent YFV 2015–2019 outbreak was driven by a viral lineage (named hereafter YFV 2015–2019) that has already circulated in the Goiás state (GO) since mid-2014 [19]. From this state of central-west Brazil, the YFV 2015–2019 lineage moved toward the densely populated southeastern states following two sylvatic dissemination routes along different primary hydrographical basins. This process originated two major YFV sub-lineages named hereafter YFVMG/SP and YFVMG/ES/RJ. The YFVMG/SP sub-lineage spread from the southwestern area of Minas Gerais state (MG) to São Paulo state (SP) along the Paraná basin. Following these events, from 2019 onward, YFV continued to spread toward south Brazil, reaching Rio Grande do Sul (RS) state, being further characterized hereafter as the YFVMG/SP/RS lineage [19,31]. The YFVMG/ES/RJ sub-lineage spread from MG state’s northern/eastern area to Espírito Santo state (ES) and then to Rio de Janeiro state (RJ), following the southeast Atlantic basin.

Nevertheless, one crucial question is whether the rapid spread of the YFV 2015–2019 lineage resulted from a complex dispersion dynamic with multiple viral transmissions between states or a few sequential founder events followed by the local establishment. Most of the studies support that YFV 2015–2019 outbreaks that occurred in these two main routes (MG/SP and MG/ES/RJ) assumed the second pattern and originated from single introductions of the virus from the southwestern and eastern region of MG, respectively [5,19,20]. Interestingly, there were two YFV 2015–2019 introductions in RJ from ES that subsequently spread along the coastal and northern sides of the Serra do Mar mountain system [19,20]. 

Here, we sequenced 43 new YFV complete genomes obtained from humans, NHPs, and, mostly, mosquitoes collected in several municipalities from RJ state during three transmission seasons (2016–2017, 2017–2018, and 2018–2019). Studying these sequences in conjunction with other YFV 2015–2019 genomes enabled us to understand better the dynamics of YFV 2015–2019 lineage spread in RJ and other states of southeastern Brazil and characterize the molecular diversity of YFV, the hosts, and vector species involved. Our study contributes to developing effective public health surveillance and prevention strategies on both regional and local scales.

## 2. Materials and Methods

### 2.1. Ethic Issues

Protocols for the capture and handling of mosquitoes and NHPs were approved by the Brazilian Ministry of Environment (SISBIO 41837-3—20 May 2015, 52472-2—26 January 2016, and 54707-5—25 August 2016) and Rio de Janeiro’s Environment Agency (INEA 012/2016—07 February 2016 and 019/2018—19 April 2018) and by the institutional Ethics Committee for Animal Experimentation (protocol CEUA/IOC-004/2015—10 April 2015, licenseL-037/2016—24 August 2016). The Ethics Committee approved the protocol for human samples for human research at the Instituto Oswaldo Cruz (IOC) (CAAE 69206217.8.0000.5248), which exempted the need for specific written informed consent from patients or their legal representatives.

### 2.2. Sampling Efforts, YFV Detection, and Sequencing Procedures

Entomological surveys were carried out in areas suspected of YFV circulation. Mosquitoes were captured using manual aspirators and entomological nets or CO_2_-baited BG-sentinel traps and processed as previously described [15,33]. Blood or liver samples were collected from dying NHP found because of alerts from an information network comprising residents from several municipalities who reported the epizooties [15,34]. The state health surveillance service provided serum samples from human patients suffering YF symptoms [19,20].

The YFV samples were obtained from 27 mosquito pools, 5 NHPs and 11 humans sampled from 26 April 2017 to 25 February 2018. All NHPs were from the species *Alouatta clamitans* (howler monkey). The mosquitoes’ samples belonged to the species *Haemagogus janthinomys*/*capricornii* (*n* = 13), *Hg. leucocelaenus* (*n* = 12), *Aedes scapularis* (*n* = 1), and *Ae. taeniorhynchus* (*n* = 1). The samples were collected in eight localities from RJ state: Macaé (*n* = 2), Maricá (*n* =16), Miguel Pereira (*n* = 1), Nova Iguaçu (*n* = 1), Angra dos Reis (continental, *n* = 1) (island, *n* = 5) territories, Teresópolis (*n* = 9), and Valença (*n* = 8) (Appendix A). 

Samples were frozen until RNA extraction with QIAmp Viral RNA Mini kit (Qiagen, Hilden, Germany). The YFV detection and viral load determination in biological specimens were performed as previously published [20]. The primers and sequencing procedures were described elsewhere [17,18,19,20].

### 2.3. Evolutionary and Phylogeographic Analyses

Previously published approaches were used to produce YFV consensus sequences and evolutionary and phylogeographic analyses using the YFV sequences generated in this study and YFV sequences available in GenBank [19]. The viral spatiotemporal spread were reconstructed using Bayesian inference with Markov-chain Monte Carlo (MCMC) sampling in the BEAST v1.10 package [35] using BEAGLE [36] to accelerate running time. The (GTR + I + Γ4) nucleotide substitution model (selected by jModelTest v1.6, [37]), a relaxed lognormal molecular clock model calibrated with a normal prior based on previous estimates [19,20], and the non-parametric Bayesian skyline coalescent model were used in case of all Bayesian phylogeographic inferences. The discrete phylogeographic reconstruction employed reversible (symmetric) and nonreversible (asymmetric) discrete phylogeographic models [38]. The continuous spatiotemporal reconstruction was estimated using geo-referenced and time-stamped sequences with a homogenous Brownian diffusion (BD) model and the heterogeneous Cauchy, gamma, and lognormal relaxed random walk (RRW) models [39]. The log marginal likelihood estimation (MLE) based on path sampling (PS) and stepping-stone sampling (SS) methods [40] was used to compare the different phylogeographic models. Bayesian analyses were run for 108 generations, and convergence (adequate sample size > 200) was inspected using TRACER v1.7 [41] after discarding 10% burn-in. The viral spatio-temporal diffusion was analyzed and visualized in SPREAD [42] and further projected in maps generated with QGIS software using public-access data collected from the Brazilian Institute of Geography and Statistics (Instituto Brasileiro de Geografia e Estatística—IBGE) [43] and National Water Agency (Agência Nacional de Águas—ANA) [44]. The mean branch dispersal velocity, change in the maximal wavefront distance from the epizootic origin, and wavefront dispersal velocity was summarized from 1000 phylogenies sampled at regular intervals from the posterior distribution (after exclusion of 10% burn-in) using the R package “seraphim” [45].

## 3. Results

### 3.1. Spatial Patterns and YFV Isolates from the 2016–2019 Outbreak in RJ State

In this work, we better tracked the YFV 2015–2019 introduction and circulation and accessed the raising of genetic diversity during the epizootic waves across RJ state. Initially, we first elucidated a set of 43 new complete YFV RJ genomes. All samples originated from both sides of the Serra do Mar mountain system at the southeast Atlantic primary river basin in the Atlantic forest biome. These areas cover two different conjugated river basins, Paraíba do Sul and Macaé, and six other tributary basins, Macaé, São João, Guandú, Guanabara Bay, Ilha Grande bay, and Paraiba do Sul (Appendix A; Figure 1). 

We aligned the 43 new YFV genomes to another 16 YFV RJ sequences retrieved from the GenBank database available in the public domain (Appendix A), from which 14 YFV sequences had previously been published by our research group [19,20,27] and two YFV sequences by Giovanetti et al., 2019 [46]. The 16 genomes were from human cases (*n* = 6), NHPs of the genera *Alouatta* (*n* = 6), and *Callithrix* (*n* = 2), *Hg. janthinomys*/*capricornii* (*n* = 1) and *Sa. chloropterus* (*n* = 1) in the 12 municipalities of RJ, Macaé (*n* = 2), Maricá (*n* = 1), Carmo (*n* = 2), Petrópolis (*n* = 1), Silva Jardim (*n* = 1), Casimiro de Abreu (*n* = 2), São Fidelis (*n* = 1), Porciúncula (*n* = 1), Guapimirim (*n* = 1), Valença (*n* = 1), Angra dos Reis in island territories (*n* = 1), and São Sebastião do Alto (*n* = 1) (Appendix A). 

The full dataset analyzed here, therefore, contains 59 YFV RJ sequences, comprising a substantial variety of samples isolated from different hosts and vectors species: *Homo sapiens* (*n* = 18), *A. clamitans* (*n* = 10), *C. jacchus*/*penicillata* (*n* = 2), *Hg. janthinomys*/*capricornii* (*n* = 14), *Hg. leucocelaenus* (*n* = 12), *Ae. scapularis* (*n* = 1), *Ae. taeniorhynchus* (*n* = 1), and *Sa. chloropterus* (*n* = 1). It encompassed 15 municipalities covering a significant and important area involved in the epidemic in southeast Brazil spanning the conjugated river basins of Paraíba do Sul (Petrópolis, Teresópolis, Valença, São Fidélis, Porciúncula, São Sebastião do Alto and Carmo,) and the Macaé conjugated river basin (Macaé, Maricá, Miguel Pereira, Nova Iguaçú, Angra dos Rei, Casimiro de Abreu, Silva Jardim, and Guapimirim) (Appendix A, respectively).

### 3.2. The Spatial Dissemination of the Sylvatic Transmission Chains YFV RJ-I and YFV RJ-II Clades 

The Bayesian discrete phylogeographic analysis confirmed that the YFV RJ sequences clustered with YFV sequences from southeastern and eastern MG, ES, and Bahia (BA) in a clade referred to as the YFV_MG/ES/RJ_ lineage (Figure 2). 

Furthermore, two introductions were successfully established where the YFV_MG/ES/RJ_ lineage diverged into two clades, YFV RJ-I and YFV RJ-II (Figure 2). The YFV_MG/SP/RS_ sub-lineage emerged in the southwest region of Minas Gerais around June 2016 (95% HPD: January 2016–October 2016) and was then disseminated once to São Paulo state around April 2017 (95% HPD: December 2016–June 2017). Similarly, the YFVMG/ES/RJ lineage emerged in the north/east region of Minas Gerais around March 2016 (95% HPD: September 2015–July 2016) and was introduced once in ES around June 2016 (95% HPD: February 2016–October 2016). From ES, the YFV_MG/ES/RJ_ sub-lineage was introduced at least six times in the Rio de Janeiro state. However, only two of these introductions were successfully disseminated throughout the state. Nevertheless, it was not possible to classify four (6.8%) YFV sequences from humans: H319/Teresópolis; H320/Teresópolis; H190/São Fidélis and H196/Porciúncula (GenBank accession numbers: MN643083, MN643084, MF538782, and MF538784, respectively) in any of the two YFV RJ clades.

Concerning the clade YFV RJ-I, we analyzed nine viral samples from *H. sapiens*, eight from *A. clamitans*, one from *C. jacchus*/*penicillata*, four from *Hg. janthinomys*/c*apricornii*, twelve from *Hg. leucocelaenus*, and one from *Ae. scapularis*, *Ae. taeniorhynchus*, and *Sa. chloropterus* each. The viral sample set of clade YFV RJ-II was composed of four samples from *H. sapiens*, two from *A. clamitans*, one from *C. jacchus*/*penicillate*, and ten from *Hg. janthinomys*/*capricornii*. The YFV RJ-I clade is formed exclusively by YFV sequences geographically widespread across nine municipalities of RJ (Macaé, Maricá, Miguel Pereira, Nova Iguaçú, Angra dos Reis, Teresópolis, Casimiro de Abreu, Silva Jardim, and Guapimirim) located essentially on the coastal side of Serra do Mar drained by the Macaé conjugated river basin. On the other hand, the YFV RJ-II clade comprised YFV sequences from the RJ municipalities of Teresópolis, Valença, Petrópolis, and Carmo, intermingled with YFV strains from neighboring sites of southeastern MG, all located in the Paraíba do Sul basin at the northern side of Serra do Mar, raising the second route of dissemination in RJ (Figure 3).

### 3.3. The YFV 2015–2019 Spread with Different Rates through the Southeast Region of Brazil but Not within the Rio de Janeiro State

We compared the rate of viral spreading of the YFVMG/ES/RJ, especially regarding the dissemination in RJ and the YFV MG/SP sub-lineages. We used a phylogenetic relaxed random walk approach to reconstruct the pattern and rate of dissemination routes of the YFV 2015–2019 lineage. The RRW model with lognormal distribution was strongly supported as the fittest diffusion model (Appendix A). The continuous phylogeographic model traced the origin of the YFV 2015–2019 outbreak to the Paraná primary river basin, which encompasses territories of GO, MG, and SP (Figure 4A). The YFVMG/SP sub-lineage spread followed the Paraná River basin, reaching the metropolitan region of SP around mid-2016. The YFVMG/ES/RJ lineage arrived in the southeast Atlantic river basin in MG at the end of 2015, reaching ES and RJ, which share this basin (Figure 4A). In RJ, the YFVMG/ES/RJ sub-lineage was further subdivided into two clades that spread following two tributary river basins: the YFV RJ-I clade dispersion followed the coastal tributary river basin, while the spread of the YFV RJ-II clade followed the Paraíba do Sul tributary river basin (Figure 4B).

We utilized a group of 146 YFV 2016–2019 genomes for our phylogeographic estimates of the epidemic wavefront through time, which indicated that the YFV 2015–2019 lineage spread up to ~900 km from its location of origin in about two years (mid-2013 to mid-2015) (Figure 4C). The median south-southeastward dispersal velocity estimated for YFV 2015–2019 was 0.48 km/day (95% HPD: 0.38, 0.59). There were little changes in the median velocity of the epidemic wavefront from mid-2013 to mid-2016, varying between 0.5 and 1.0 km/day. Between 2017 and 2018, there was a slight increase above 1.0 km/day, followed by a drastic fall in 2019 (Figure 4D).

We found that the dispersal velocity of the YFVMG/ES/RJ lineage [0.69 (95% HPD: 0.52, 0.85)], which followed the southeast Atlantic primary river basin, was significantly higher than that of the YFVMG/SP [0.21 (95% HPD: 0.11, 0.31)], which dispersed through the Paraná primary river basin (Figure 5). We also detected lower dispersal velocities of YFV in SP and ES (medians: 0.21 and 0.23 km/day, respectively) when compared with MG (median: 0.54 km/day) and RJ (median: ≥0.43 km/day) states. Curiously, despite differences in characteristics, such as predominant altitude, forest fragmentation, and human population density, between the coastal and continental portions separated by the Serra do Mar, no significant differences were found between the median dispersal velocities of the YFV along the two sylvatic transmission routes in RJ (Figure 5).

The two routes of YFV dissemination in RJ state arose on each side of Serra do Mar, a 1500 km long system of mountain chains and escarpments extending from ES to RS. As observed in Serra do Mar of RJ, the escarpment forms the boundary between the sea-level shore and the inland plateau. Several of the municipalities touched by the epizootic wave are in the upper part of Serra do Mar, and their territories encompass areas in both the coastal and continental slopes of this mountain chain. Our study detected only one of the two transmission chains (YFV RJ-I or YFV RJ-II) in each sampled municipality. This finding is consistent either by analyzing numerous viral samples from a single location (e.g., Maricá, Valença, or Angra dos Reis) or samples from different hosts from the same area (e.g., Maricá, Casimiro de Abreu, Macaé, and Angra dos Reis).

These two successful intra-RJ lineages, the YFV RJ-I and YFV RJ-II clades, comprise 62% and 29% of all sequences from the state, respectively, and all YFV sequences were recovered from NHPs and mosquitoes from Rio de Janeiro, with only one exception. We also detected two independent introductions of the YFV RJ-II clade from RJ into the southeastern region of Minas Gerais. This analysis supports those geographic barriers; notably, mountain chains played a significant role in the YFV sylvatic dissemination through the southeast region of Brazil. Most YFV cases detected in different states reflect the local establishment of one or a few sylvatic transmission chains.

Interestingly, the only exception was the municipality of Teresópolis, where the virus sampled in 2018 from humans and mosquitoes clustered in both the coastal (YFVRJ-I; strains H313 and H317) and continental (YFV RJ-II; strains H295, H299, H312, H300, and TR2807) transmission chains. (Appendix A, Figure 2). Moreover, two virus genomes (H319 and H320) clustered outside the RJ YFV clades and were basally associated with the continental chain clustered in a distinct basal sub-clade with samples of a previous transmission season (early 2017) from localities in ES and northern RJ, quite distant from Teresópolis (Figure 2). In summary, there is evidence that at least three distinct YFVs were introduced into Teresópolis in a single transmission season (December 2017 to February 2018), which co-circulated in an area of approximately 38 km^2^ (Figure 6).

### 3.4. Mapping the Genetic Diversity Raised in YFV 2016–2019 Outbreak in RJ State

To characterize the genetic diversity of the YFV outbreak in RJ and gain more insights into its evolution, we sequenced the genome of the 43 newly sampled YFV and compared them with 16 YFV RJ sequences retrieved from databases available in the public domain (Figure 2).

We observed that 54% (32/59) of the YFV genomes analyzed had 25 divergences at the amino acid level along the polyprotein involving both structural proteins and non-structural proteins (NS) concerning the reference YFV GO27 genome (Genbank accession No. MK333804) of this study, isolated in 2015 during the 2015–2019 epidemic wave.

Amino acid variations were observed in the capsid C (V79A), (R82K), (Q103R), (I1086V); prM (L131S), (A254V); envelope E (T526A); NS1 (A821S), (A826S), (E829D) (F978L), (T993I), (I1086V); NS2A (S1313L); NS2B (E1417G); NS3 (Y1503H), (E1744G), (E1831G); NS4A (I2176V); NS4B (K2502E) and NS5 (I2644T), (I2764V), (N2897K), (I3128M); (T3151I).

In clade RJ-I, 15 out of 36 YFV genomes showed 16 amino acid substitutions, 1 single amino acid change in 13 genomes, 1 double amino acid change in one genome, and 6 changes in the most recent sample, isolated in 2019 from NHP. The NS1/F978L change was found in YFV from 4 mosquito pools: 2 *Hg. leucocelaenus*, 1 *Ae. scapularis*, and 1 *Ae. taeniorhynchus*.

We found only two variations between 17 genomes of the clade YFV RJ-II, both at NS: NS1 and NS4A. The NS4A/I2176V, the most frequent variation detected in this study (18/25), was found in all YFV genomes regardless of the host from which they were sampled (H, NHP, and M) and was the unique change in common to both YFV RJ clades. The NS1/I1086V change was exclusively observed in *Hg. janthinomys*/*capricornii* mosquitoes captured at the same RJ municipality, Valença. These viral samples also bear the change NS4A/I2176V.

## 4. Discussion

Since the 2000s, Brazil has experienced the expansion of the YF transmission area of enzootic/sylvatic circulation, giving rise to an outbreak with seasonal transmission waves from 2015 to 2021 [1,2,6]. The most affected region, in the coastal states of the southeast, shares one of the richest biomes, the Atlantic Forest, which has been considered YFV-free and without vaccine recommendation for decades [2,11,15]. RJ state is completely inserted in the Atlantic Forest, interspersed with forest remnant fragments of variable extents, stretching from the coast to the inland mountains, with high biodiversity and diversity of altitudes, temperature regimes, and rainfall. RJ is divided by a long mountain chain segregating the two main primary rived basins, the Paraíba do Sul conjugated basin and Macaé conjugated basin [26,29].

The objective of this study was to analyze YFV RJ sequences obtained from the high number and variety of the host and vector species circulating in such a singular and variable ecological context. The study of molecular and phylogenetic aspects of YFV spread in virgin zones on a more local scale constitutes a unique opportunity to infer the dynamic dispersion and the accumulation of genetic divergence. The new 43 sequenced YFV genomes from different hosts were analyzed composing a dataset of 59 YFV RJ genomes from 15 municipalities. It consists of a substantial variety of samples isolated from different hosts and vector species implicated in a single outbreak.

In our study, we confirmed that the YFV RJ sequences clustered with YFV sequences from southeastern and eastern MG, ES, and BA, in a clade referred to as YFVMG/ES/RJ lineage, and toward two successful introductions from ES in RJ diverging into clades YFV RJ-I and YFV RJ-II, respectively, along the coastal and northern sides of the Serra do Mar mountain system [19,20], supporting previously published studies. 

A more recent study, by contrast, suggests that YFV 2015–2019 infections in RJ resulted from multiple introductions from ES over time [47]. This last study, however, ignored whether multiple introductions resulted from human cases that may have acquired the infections outside RJ and/or the existence of several sylvatic transmission chains within RJ. Nevertheless, despite the magnitude of this YFV outbreak, there is almost no data on the number of viral introductions and the dynamic of transmission chains in a smaller geographic location like RJ [48].

Other controversial findings were reported about the rate of geographical dispersal of the YFV 2015–2019 lineage. Early studies estimated that the YFV 2015–2019 lineage moved, on average, 3.0–4.3 km/day during the wet and warmer months (January–April) of 2017 [5,9,20]. Later studies covering multiple epidemic seasons, however, provide more conservative estimates. Using viral sequences from the YFVMG/ES/RJ and YFVMG/SP lineages, Delatorre et al. (2019) [19] estimated that the virus moved on average 0.48 km/day, whereas Giovanetti et al. (2019) [46] estimated an average dispersal rate of 0.12 km/day for the lineage YFVMG/ES/RJ, and Hill et al. (2020) [9] estimated a mean dispersal rate of 0.83 km/day for the YFVMG/SP lineage. This variation in the rate of YFV spread might be due to different datasets analyzed between the studies, which spanned different temporal intervals and involved mainly diversity and amount sampling of NHP, and mosquito vectors and the distances typically traveled by them in these fragmented forest areas during the sylvatic cycle [5,9,47].

A well-defined dispersion of two transmission chains, YFV RJ-I or YFV RJ-II, in each sampled municipality was detected in our study. The clade YFV RJ-I was formed by numerous viral samples from a single location propagating by the coastal chain to RJ along the Macaé conjugated basin. The clade YFV RJ-II, represented by samples from different hosts from the same area, disseminated in the continental slopes of the Serra do Mar. These two independent clades demonstrate the importance of natural barriers (such as mountains chains) for limiting viral dispersion by restraining the movement of vectors, NHPs, and, consequently, the spread of YFV. While the spread of RJ-II was apparently limited to the southern border of RJ in April 2018, RJ-I continued to expand southward along the seashore reaching latitude 23°40′ at the north coast of SP (Caraguatatuba), in July 2018, as indicated by the analysis of recently published viral genome sequences (GenBank ON022287, ON022285, ON022543, ON022544, and ON022660 [49]).

Recently, mathematical models have shown that human-modified areas such as large cities, pastures, and forestry limit viral spread [50,51]. Based on our phylogenetic results, it is possible to include mountain chains as a limiting factor. Despite the independence of the YFV RJ-I and YFV RJ-II clades and the geographic diversity across the two dissemination routes, the speed of viral dispersion was similar. In the southeast, the highest YFV dispersal speeds were associated with a landscape composed of interconnected forest fragments surrounded by pastures and/or roads, which generate high edge densities [50,51,52]. This type of matrix intensifies the dispersion of infected primatophilic *Haemagogus* mosquitoes between fragments through flight stimulated by the search of available breeding sites and blood sources following devasting NHP epizootics or wind currents. It increases the chance of contact with humans and susceptible NHPs [50,51,53,54]. This same type of matrix can be found throughout RJ, including the coastal and continental slopes of Serra do Mar, which may explain the similarity in dispersion speed. In addition, mosquito sampling in forest fragments on both slopes showed the same pattern: high YFV infection rates associated with low species richness and a high abundance of *Haemagogus* mosquitoes [28].

The exception to the geographic isolation of the YFV RJ-I and YFV RJ-II clades was the municipality of Teresópolis, where a peculiar clustering occurred. The 2018 YFV genomes from humans and mosquitoes from Teresópolis grouped in the YFV RJ-I, YFV RJ-II, and ES YFV clusters, corresponding to the basal sub-clade of the YFV, close to samples of a previous transmission season. Interestingly, we detected two genomes from Teresópolis clustering in the YFV RJ-I clade, which was exclusively detected on the coastal side of Serra do Mar. These YFV genomes were detected in people living on the continental slope of Teresópolis, but very close to the watershed between the Macaé (coastal) and Paraíba do Sul (continental) conjugated river basins (Figure 6). Teresópolis is a very popular mountain touristic municipality with human settlements surrounded by conserved forested areas, crossed by important highways. This triple YFV cocirculation recorded in Teresópolis in a single transmission season may be related to human movements, one of the factors inducing the YFV spatial spread [55], but also, highways lined by forests like those abundantly crossing Teresópolis could have served as dispersal routes for infected sylvatic mosquitoes through wind tunnels and helped them to overcome the watershed separating the two river basins [48]. This circumstance was detected by the analysis of a considerable number of samples collected in the same outbreak on small scales as in the present study.

Entomological and epizootic surveillance must be constant in the SYF context. Several studies have described the genetic and phylogeographic patterns involving NHP epizootics and human cases in the recent outbreak [5,9,17,18,19,20,28,31,46,47,48]. In this study, however, a notable contribution is the unprecedented analysis of 29 YFV genomes from mosquito species belonging to five species of three genera.

Entomological and ecological surveys conducted before and during the outbreak in the Brazilian states that affected YFV described the role of these vectors’ species in the sylvatic YF outbreak. Briefly, *Hg*. *janthinomys/capricornii* and *Hg. leucocelaenus* were considered primary vectors because of density, abundance, distribution, and infection rates and were the source of 48 and 41% of analyzed YFV genomes, respectively [15]. In contrast, *Sa. chloropterus*, *Ae. scapularis*, and *Ae. taeniorhynchus*, considered secondary vectors or opportunistic mosquitoes [15,28], contributed to only 1% of YFV samples.

The high infection rates in the species of *Haemagogus* were associated with ecological characteristics commonly found in forest fragments, such as lower species richness, higher abundance of these mosquitoes, and the lower normalized difference vegetation index (NDVI) [15,28]. The large number of positive pools in Maricá (*n* = 16; coastal) and Valença (*n* = 8; continental) illustrates this pattern well. In both locations, the dominance of *Haemagogus* mosquitoes was noticed in the forest fragments, surrounded by pastures, farms, and roads, enhancing viral dispersion. Interestingly, we observed remarkable clustering genomic variations in mosquito samples from RJ. The NS1/F978L was only found in the YFV from *Hg. leucocelaenus*, *Ae*. *scapularis*, and *Ae. taeniorhynchus* captured in Maricá (coastal). Moreover, the NS1/I1086V change was exclusively observed in *Hg. janthinomys*/*capricornii* from Valença (continental), being absent in all *Hg. leucocelaenus* simultaneously obtained in the same area, but, curiously, these *Hg. janthinomys*/*capricornii* YFV genomes from Valença additionally carried the change in NS4A/I2176V shared by all sequences in clade YFV RJ-II in the YFV isolated from humans and NHPs from other sites.

NS1 is a multi-functional protein that plays roles in virus replication, pathogenesis, and the immune response to virus infection [56]. Dissemination of YFV has been accompanied by adaptive genetic diversifications, promoting escape from both mosquito and mammal antiviral responses. Current interest in developing recombinant flaviviral vaccines incorporating NS1 prompts additional study of the basis for protection conferred by antibodies to this protein and the evolution pattern following the successive outbreaks [55,56]. YFV GO27, the oldest sequence of the analyzed outbreak, displays the nine unique amino acid signatures characteristic of the 2016–2019 YFV southeastern transmission. This molecular signature characterizes the founding strain and the other YFV that followed the outbreak analyzed in this study, and an essential role of this signature in viral fitness in invertebrate and vertebrate hosts has been demonstrated [17,19]. We observed that 54% (32/59) of the YFV genomes analyzed had 25 divergences at the amino acid level along the polyprotein relative to the YFV GO27 genome isolated from a capuchin *Sapajus libidinosus* in 2015 from GO, west-central region. The availability of the complete genomic YFV sequences may facilitate the further phenotypic evaluation of viral differences. The potential significance of variable areas has been used in wild YFV isolates and infectious clone studies to improve understanding of the potential impact on viral fitness, transmissibility, vector competence, and virulence. With such approaches, it will be possible to infer vaccine efficiency since, given the re-emergence of YF outbreaks, health authorities have implemented vaccination campaigns as the sylvatic cycle expands to YF-free areas. Using molecular and phylogenetic analysis, combined with the phylogeography of YFV dispersion in well-characterized environments in terms of ecology and host diversity, we describe the epidemiological impact of YFV outbreaks in the state of RJ.

In addition to the epizootic cycles, which follow a more traceable route of emergence and transmission, the maintenance of the sylvatic cycle with great diversity and abundance of vectors observed in RJ, with the risk of YF re-urbanization, is highly concerning. For this reason, entomological surveillance, with a description of vector species carrying YFV, associated with genomic sequencing of this virus and analyses such as those presented here, constitute an essential tool to fill gaps in knowledge about the persistence of YFV in fragment forests and predict regions of vulnerability to transmission avoiding spillovers to YFV-free areas. The dispersal routes inside RJ identified in this work represent a perspective regarding the surveillance of YFV in Brazil.

## Figures and Tables

**Figure 1 viruses-15-00437-f001:**
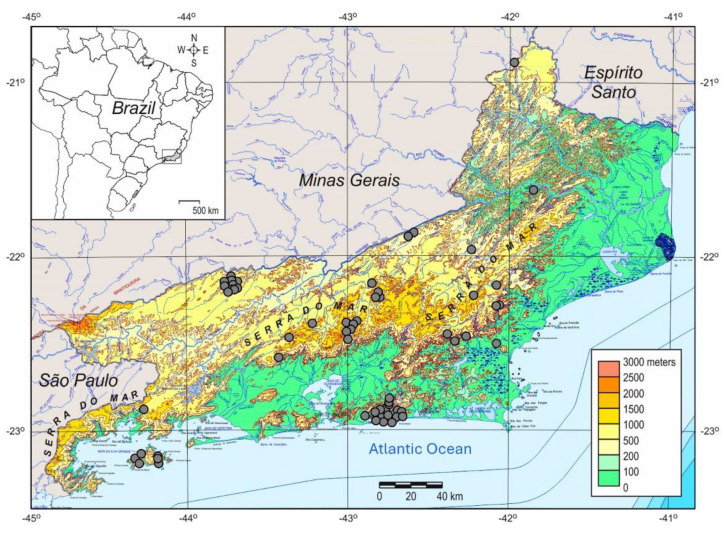
Rio de Janeiro (RJ) state topographic map depicting the Serra do Mar, a system of mountain chains and escarpments, and the coastal lowland regions, and the neighboring states of Minas Gerais, Espírito Santo, and São Paulo, comprehending southeastern Brazil. The gray circles indicate sample collection location in the RJ state.

**Figure 2 viruses-15-00437-f002:**
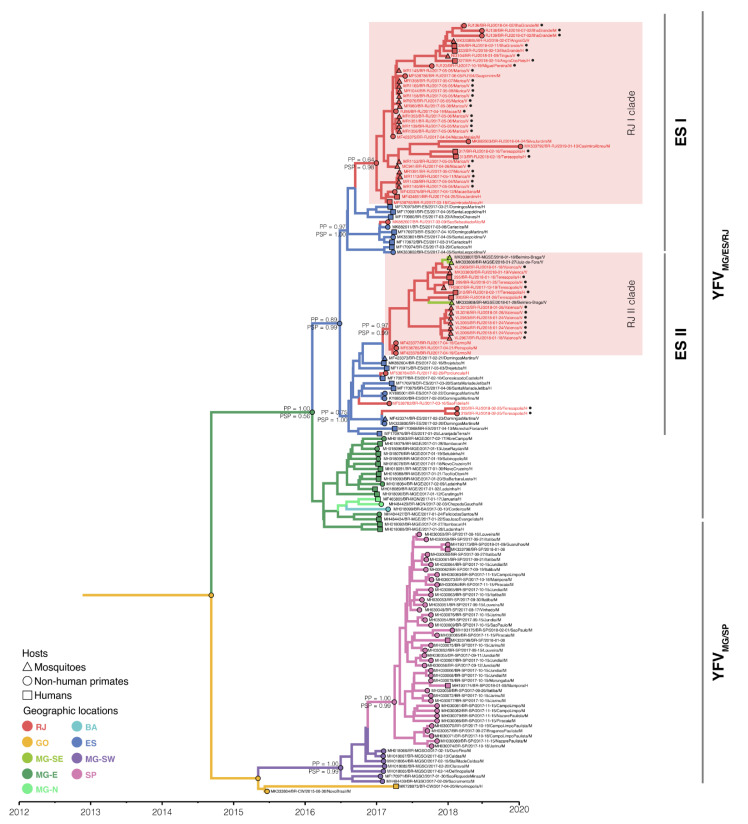
Time-scaled maximum clade credibility phylogeny of the YFVMG/ES/RJ lineage. The branches’ colors represent the most probable locations of their descendent nodes, as indicated in the legend. Tips names were codified as accession number/date/region/host, and tip symbols indicate the host as indicated in the legend. The tip labels corresponding to the data set of this study, 43 RJ YFV samples sequenced in this study (black dots in front of sequence names), and 16 RJ YFV from the GenBank database are colored red. All horizontal branch lengths are drawn to a scale of years. The location refers to Brazilian (BR-) following the state or region of the country: RJ, Rio de Janeiro; CW, central-western; BA, Bahia; MG-SE, southeastern Minas Gerais; MG-N, northern Minas Gerais; MG-E, eastern Minas Gerais; ES, Espírito Santo; MG-SO, southwest Minas Gerais; SP, São Paulo.

**Figure 3 viruses-15-00437-f003:**
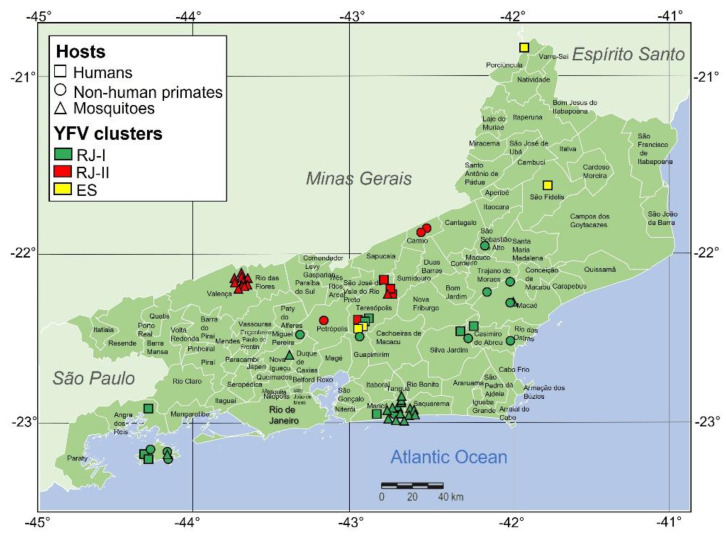
Spatial distribution of YFV samples of the 2017–2019 outbreak from 15 Rio de Janeiro state municipalities. YFV sequences form the YFV RJ-I clade from sites drained by the Macaé conjugated river basin (Macaé, Maricá, Miguel Pereira, Nova Iguaçú, Angra dos Reis, Teresópolis, Casimiro de Abreu, Silva Jardim, and Guapimirim). The YFV RJ-II clade, covering the conjugated river basins Paraíba do Sul, is formed by YFV sequences from Petrópolis, Teresópolis, Valença, and Carmo.

**Figure 4 viruses-15-00437-f004:**
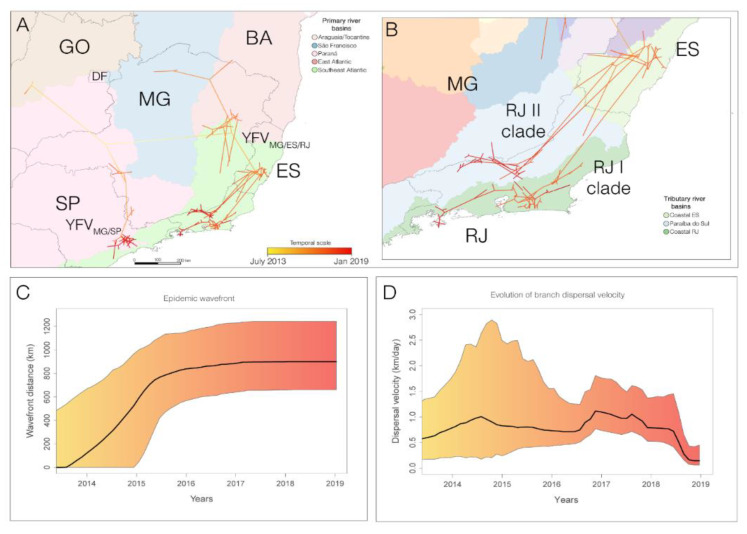
Reconstructed spatiotemporal diffusion of the YFV lineage from central-west, Goiás state, and toward southeast Brazil. Phylogeny branches were arranged in space according to the internal nodes’ locations inferred by the continuous phylogeographic model. Branches were colored according to time, as indicated by the legend. The dark gray lines represent the Brazilian state’s boundaries while the colored areas show the different primary river basins (**A**) and additional details on tributary basins (**B**), where are depicting the two routes of YFV of dissemination in RJ state arose on each side of Serra do Mar, expanding to clade YFV RJ-I and clade YFV RJ-II. Progression of maximal wavefront distance (**C**) estimated for the YFV 2015–2019 outbreak based on continuous phylogeographic analysis. The wavefront represents the spatial distance between the farthest extent of the wavefront and the position of the most ancestral node. Progression of mean dispersal velocity over time (km per day) (**D**) estimated for the YFV 2015–2019 outbreak. For (**C**,**D**), the colored area corresponds to the 95% credible regions of the estimation. The Brazilian states boundaries and primary river basins information were retrieved from Instituto Brasileiro de Geografia e Estatística [IBGE] [43]. Brazilian states: DF, Distrito Federal; ES, Espírito Santo; GO, Goiás; MG, Minas Gerais; RJ, Rio de Janeiro; SP, São Paulo.

**Figure 5 viruses-15-00437-f005:**
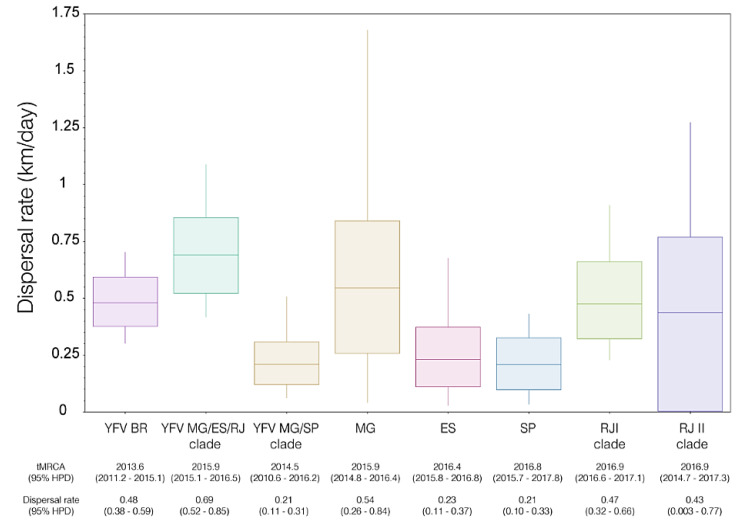
Inter- and intrastate estimates of the viral dispersal rate of the YFV 2015–2019 outbreak in southeast Brazil. The box plots represent the median dispersal rate (km per day) and the 95% HPD intervals of the posterior distributions estimated under the lognormal RRW continuous phylogeographic model. The tMRCA and dispersal rate values (the HPD are in the parenthesis) are shown under each boxplot. YFV BR—all datasets; Brazilian states: ES, Espírito Santo; MG, Minas Gerais; RJ, Rio de Janeiro; SP, São Paulo.

**Figure 6 viruses-15-00437-f006:**
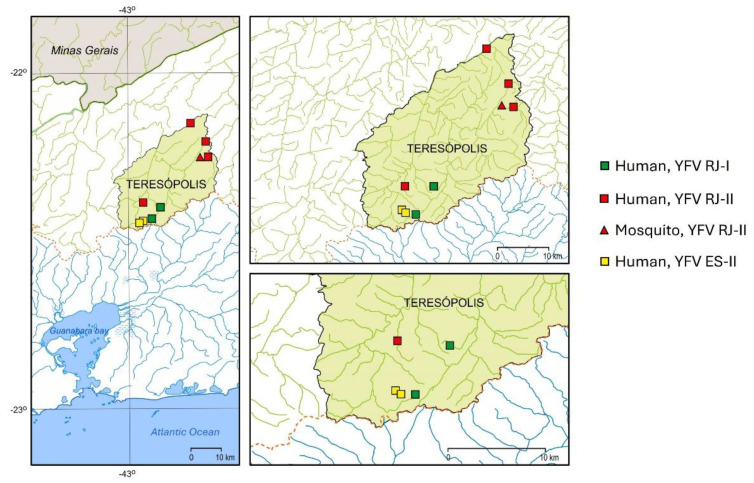
Spatial distribution of YFV samples from humans (square) and mosquitoes (triangle) from Teresópolis municipality. YFV samples clustered in clades YFV RJ I (green; coastal), YFV RJ-II (red; continental), and YFV ES II (yellow). The tributary rivers of the Macaé (coastal) and Paraíba do Sul (continental) conjugated river basins are outlined in green and blue, respectively.

## Data Availability

Not applicable.

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
