# Peer review of "Ecological, Genetic, and Phylogenetic Aspects of YFV 2017–2019 Spread in Rio de Janeiro State"

_viruses, 2023, doi:10.3390/v15020437_

Round 1
Reviewer 1 Report
Ribeiro et al present research describing the molecular relatedness of YFV isolates from the recent 2017 -2019 transmission in Brazil. The key points of the MS include the independent evolution and spread of the two intra RJ lineages (RJ-I and RJ-II) that appear to have diverged following independent introduction events on either side of the Serra do Mar, an escarpment that forms a barrier between the coastal region and and the inland plateau of the state.
The work will be of interest to scientists who study flavivirus evolution and the ecology of flaviviruses, especially yellow fever virus. Overall the work is signficant and adds to our understanding of contemporary YFV transmission in Brazil. My overarching critique is that the MS is too lengthy as submitted, both the introduction and discussion would benefit from being substantially shorter and more concise.
Comments:
Larger figures needed throughout.
Figure 1 and Figure 3 can be combined. The geography and source of virus sequence (NHP, human, mosquito) can reasonably be shown in a single map.
Figure 2 should have the figures for hosts (square for human, circle, triangle) placed alongside each of the 43 new isolates sequenced so that the evolutionary relationship between the viral genomes can also be evaluated with respect to what host the virus was in. Figure 2 is really too small to adequately evaluate the isolates on the tree.
The four sequences from humans that did not classify with the other viruses should be specifically identified with a star or other on the tree and in the combined figure 1+3.
Figure 4 was too small for me to interpret. Noe that in the figure legend frames A and B are described but frames C and D are not.
Two viruses isolated in Teresopolis stand apart, genetically, from the other viruses characterized, H319 and H320. They should be identified with a marker on Figure 2.
Figure 6 needs a key for the colors and shapes for quick reference.
The discussion is too lengthy. All text that is first presented in the introduction should be cut from the discussion. Many of the results are fully restated in the discussion section, I suggest the these statements be reduced to short summaries of the key results pertaining to each discussion point.
The Teresopolis discussion does a reasonable job presenting hypotheses explaining how the divergent or genetically distinct viruses may have been detected there - it seems less likely that human movement (infected humans moving from one place to another) would have resulted in two humans infected somewhere else with the same strain coming to Teresopolis and becoming ill. It should be noted that no mosquitoes or NHP samples were collected in the south of Teresopolis. Is it possible to identify the location and species for the other ES strains in the phylogenetic tree. Have mosquitoes or NHPs with the ES strain been identified in this municipality.
The difference in the number of coding mutations between RJ-I and RJ-II is striking. Tying these mutations to the mosquito spp they were isolated from is highly speculative and lacks specific mechanistic hypotheses. That said, there appear to have been rather distinct selective pressures acting on the two different isolate populations. An analyses of synonymous to non-synonymous mutation rates between the two clades may provide insight into whether the pressures are different.
Minor comment
Mosquito spp should be in italics
Author Response
(i) Reply for Reviewer 1
Thank you very much for your comments.
“Ribeiro et al. present research describing the molecular relatedness of YFV isolates from the recent 2017 -2019 transmission in Brazil. The key points of the M.S. include the independent evolution and spread of the two intra R.J. lineages (RJ-I and RJ-II) that appear to have diverged following independent introduction events on either side of the Serra do Mar, an escarpment that forms a barrier between the coastal region and and the inland plateau of the state.
The work will be of interest to scientists who study flavivirus evolution and the ecology of flaviviruses, especially yellow fever virus. Overall the work is signficant and adds to our understanding of contemporary YFV transmission in Brazil. My overarching critique is that the M.S. is too lengthy as submitted, both the introduction and discussion would benefit from being substantially shorter and more concise.”
Comments:
Larger figures needed throughout. Figure 1 and Figure 3 can be combined. The geography and source of virus sequence (NHP, human, mosquito) can reasonably be shown in a single map.
Reply:
We chose to leave the two figures separated to provide more fluent information to the reader. As you can see, Figure 1 is a relief map showing the mountain chains and basins, so the reader can see where the samples were collected, associating them with the R.J. topography. Figure 2 shows the establishment of the phylogenetic tree determining the classification of each genome in cluster RJ I or RJII. Figure 3 shows the R.J. municipalities, the host source, and the clade where the samples belong, exhibiting the two main routes of the YFV outbreak. In our opinion, the flow of data facilitates the reader’s understanding.
Figure 2 should have the figures for hosts (square for human, circle, triangle) placed alongside each of the 43 new isolates sequenced so that the evolutionary relationship between the viral genomes can also be evaluated with respect to what host the virus was in. Figure 2 is really too small to adequately evaluate the isolates on the tree.
Reply:
We modified Figure 2 to a better resolution of the letters. We also inserted a black dot indicating the new 43 YFV genomes obtained in our work. We included a legend describing the hosts, as suggested by the reviewer. The new version of this figure made Supplementary Figure 1 unnecessary.
The four sequences from humans that did not classify with the other viruses should be specifically identified with a star or other on the tree and in the combined figure 1+3.
Reply:
In figure 3, these genomes are identified as colored yellow, two in Teresópolis, 1 in São Fidelis, and one in Porciuncula.
Figure 4 was too small for me to interpret. Note that in the figure legend, frames A and B are described but frames C and D are not.
Reply:
Thank you for detecting the mistake in the legend. We included the description of frames C and D.
Two viruses isolated in Teresopolis stand apart, genetically, from the other viruses characterized, H319 and H320. They should be identified with a marker in Figure 2.
Reply:
We have improved the resolution of figure 2 and labeled the new 43 YFV genomes, allowing a proper identification of the new viral genomes in the tree. So, the H319 and H320 genomes are without difficulty identified because they are out of the clade RJII grouping inside ES clade.
Figure 6 needs a key for the colors and shapes for quick reference.
Reply:
Figure 6 was amended as suggested by referee 1.
“The discussion is too lengthy. All text that is first presented in the introduction should be cut from the discussion. Many of the results are fully restated in the discussion section, I suggest these statements be reduced to short summaries of the key results pertaining to each discussion point. “
Reply:
The section Discussion was revised and partially rewritten.
“The Teresopolis discussion does a reasonable job presenting hypotheses explaining how the divergent or genetically distinct viruses may have been detected there - it seems less likely that human movement (infected humans moving from one place to another) would have resulted in two humans infected somewhere else with the same strain coming to Teresopolis and becoming ill. It should be noted that no mosquitoes or NHP samples were collected in the south of Teresopolis. Is it possible to identify the location and species for the other E.S. strains in the phylogenetic tree. Have mosquitoes or NHPs with the E.S. strain been identified in this municipality.”
Reply:
Thank you for your comments and questions. It is indeed possible to identify the location and species of the other E.S. lineages in the phylogenetic tree of Figure 2. E.S. samples are on the darkish blue branches, the sample name includes the name of the municipality and it is possible to know if the virus was obtained from vector (V), monkey (M) or human (H). For example, the figure shows the two human samples from Teresópolis grouped into a clade with virus obtained from from mosquito and in monkeys from E.S. (municipality of Domingos Martins). Regarding the second question, unfortunately there are no available genome sequence of yellow fever virus obtained from non-human primates from Teresópolis, either in our sampling or in the GenBank. And the only virus obtained from mosquitoes in this municipality is grouped in the RJ-II clade (Figure 2). Moreover, the virus obtained from monkeys from the municipalities closest to Teresópolis did not cluster with those from E.S., but with RJ-I (Guapimirim) and RJ-II (Petrópolis) (Figs 2 and 3).
The difference in the number of coding mutations between RJ-I and RJ-II is striking. Tying these mutations to the mosquito spp they were isolated from is highly speculative and lacks specific mechanistic hypotheses. That said, there appear to have been rather distinct selective pressures acting on the two different isolate populations. An analysis of synonymous to non-synonymous mutation rates between the two clades may provide insight into whether the pressures differ.
Reply:
It is an exciting point of discussion and requires a very complex approach to get some knowledge of the genetic composition of the viral population in different habitats and species of host populations can influence the frequency of mosquito populations allowing higher viral proliferation rates. Consequently, a higher number of coding mutations would be expected or not. It is not our goal in the study. However, we will get some insight into this in another study using a dataset composed of the Brazilian YFV genomes bound to the recent outbreak.
Minor comment
Mosquito spp should be in italics
Reply:
The text was amended throughout.
Reviewer 2 Report
The manuscript is very interesting and provides a lot of information.
some points to be addressed
1-The review needs linguistic editing since the sentence sometimes is long.
2- Some comments should be considered, the comment is found in the attached manuscript as sticky notes.

Author Response
The manuscript is very interesting and provides a lot of information.
some points to be addressed
Reply:
Thank you for your detailed and helpful revision.
1-The review needs linguistic editing since the sentence sometimes is long.
Reply:
We corrected the text’s spelling and modified some parts to reduce the sentences’ size to increase the work’s understanding.
2- Some comments should be considered, the comment is found in the attached manuscript as sticky notes.
Reply:
We amended the text as recommended in the sticky notes. All modifications are marked.
In material and methods:
- We provided the manufacturer and the kit used in viral RNA extraction.
- The methods/primers used for genome amplification and sequencing were detailed and described in previously published articles, as cited in our paper:
Bonaldo, M.C.; Gomez, M.M.; Dos Santos, A.A.; Abreu, F.V.S.; Ferreira-de-Brito, A.; Miranda, R.M.; Castro, M.G.; Lourenco-deOliveira, R. Genome analysis of yellow fever virus of the ongoing outbreak in Brazil reveals polymorphisms. Mem. Inst. Oswaldo Cruz 2017, 112, 447-451, doi:10.1590/0074-02760170134.
Barbosa, C.M.; Di Paola, N.; Cunha, M.P.; Rodrigues-Jesus, M.J.; Araujo, D.B.; Silveira, V.B.; Leal, F.B.; Mesquita, F.S.; Botosso, V.F.; Zanotto, P.M.A.; Durigon, E.L.; Silva, M.V.; Oliveira, D.B.L. Yellow Fever Virus RNA in Urine and Semen of Convalescent Patient, Brazil. Emerg. Infect. Dis. 2018, 24, 1, 176–8. doi: 10.3201/eid2401.171310. Epub 2018 Jan 17. PMID: 29058663; PMCID: PMC5749440.
Gomez, M.M.; Abreu, F.V.S.; Santos, A.; Mello, I.S.; Santos, M.P.; Ribeiro, I.P.; Ferreira-de-Brito, A.; Miranda, R.M.; Castro, M.G.; Ribeiro, M.S.; et al. Genomic and structural features of the yellow fever virus from the 2016-2017 Brazilian outbreak. J. Gen. Virol. 2018, 99, 536-548, doi:10.1099/jgv.0.001033.
Delatorre, E.; de Abreu, F.V.S.; Ribeiro, I.P.; Gomez, M.M.; Dos Santos, A.A.C.; Ferreira-de-Brito, A.; Neves, M.; Bonelly, I.; de Miranda, R.M.; Furtado, N.D.; et al. Distinct YFV Lineages Co-circulated in the Central-Western and Southeastern Brazilian Regions From 2015 to 2018. Front. Microbiol. 2019, 10, 1079, doi:10.3389/fmicb.2019.01079.
The software and the parameters used for the phylogenetics are described in the same paragraph, where we detail the adaptations from a previous analysis employed in this paper:
Delatorre, E.; de Abreu, F.V.S.; Ribeiro, I.P.; Gomez, M.M.; Dos Santos, A.A.C.; Ferreira-de-Brito, A.; Neves, M.; Bonelly, I.; de Miranda, R.M.; Furtado, N.D.; et al. Distinct YFV Lineages Co-circulated in the Central-Western and Southeastern Brazilian Regions From 2015 to 2018. Front. Microbiol. 2019, 10, 1079, doi:10.3389/fmicb.2019.01079.
In the results section:
- We transferred the second paragraph to the material and methods section 2.2.
- We altered Figure 2.
In the discussion:
- We added a final phrase to the paragraph regarding the impact of our results, as follows:” The dispersal routes inside R.J. identified in this work represent a piece of perspective regarding the surveillance of YFV in Brazil.”
Reviewer 3 Report
This is a very interesting article that evaluates aspects of the ecology, genetics and dispersion of the yellow fever virus in the Atlantic Forest Biome, in one of the most populous states in Brazil. It represents yet another work that adds to a series of others related to the epidemic from 2013 to 2022 in the country, and for that reason alone, it could already be considered relevant. It is worth mentioning the fact that it is an analysis on a sub-regional to local geographic scale, which brings not only more specific conclusions of scientific interest, but also a valuable set of discussions that will serve as a reference for technicians and managers of epidemiological surveillance. Another highlighted aspect was the inclusion of dispersion speed data from other regions for a comparative analysis, which was very opportune. Well, in this way, I congratulate the authors, and inserted a few simple notes:
- in the second paragraph of the introduction, when the authors mention the susceptibility of nonhuman primates, it is not essential, but perhaps it is worth mentioning one of the references:
de Azevedo Fernandes NCC, Guerra JM, Díaz-Delgado J, et al. Differential Yellow Fever Susceptibility in New World Nonhuman Primates, Comparison with Humans, and Implications for Surveillance. Emerg Infect Dis. 2021 Jan;27(1):47-56. doi: 10.3201/eid2701.191220. PMID: 33350931; PMCID: PMC7774563.
Mares-Guia, M.A.M.d., Horta, M.A., Romano, A. et al. Yellow fever epizootics in non-human primates, Southeast and Northeast Brazil (2017 and 2018). Parasites Vectors 13, 90 (2020). https://doi.org/10.1186/s13071-020-3966-x
- in the third paragraph on page 5, correct to “Sa.” (Sabethes);
- at the end of the second paragraph on page 6, are there any details about the epidemiological investigation/confirmation of autochthony of the 4 human cases whose genotypes could not be classified into any of the clades? For the cases of Teresópolis, as mentioned in the discussion, could perhaps be related to the proximity of Maricá and/or high circulation of visitors, but regarding the other two cases is there any information? Well, the dates of the cases are inserted in a very tight sequence with the dates of the surrounding cases. Anyway, intriguing.
- regarding the results on comparisons between dispersion speeds, in general, for the reader who is not a specialist in statistics (as is my case), the medians seem to satisfactorily describe the differences between each region, especially when we note the range of variation . But something I missed a bit was graphically visualizing the speed variation within the two periods, from February to June 2017 and from October 2017 to April 2018, according to supplementary tables 1 and 2.
- in the discussion, in the last paragraph on page 11 the authors referred to “smaller geographic scales”; wouldn't they be bigger geographical scales? For example 1:50,000>1:500,000. The same can be seen at the beginning of page 13 under “lower scales”.
Author Response
This is a very interesting article that evaluates aspects of the ecology, genetics and dispersion of the yellow fever virus in the Atlantic Forest Biome, in one of the most populous states in Brazil. It represents yet another work that adds to a series of others related to the epidemic from 2013 to 2022 in the country, and for that reason alone, it could already be considered relevant. It is worth mentioning the fact that it is an analysis on a sub-regional to local geographic scale, which brings not only more specific conclusions of scientific interest, but also a valuable set of discussions that will serve as a reference for technicians and managers of epidemiological surveillance. Another highlighted aspect was the inclusion of dispersion speed data from other regions for a comparative analysis, which was very opportune. Well, in this way, I congratulate the authors, and inserted a few simple notes:
- in the second paragraph of the introduction, when the authors mention the susceptibility of nonhuman primates, it is not essential, but perhaps it is worth mentioning one of the references:
de Azevedo Fernandes NCC, Guerra JM, Díaz-Delgado J, et al. Differential Yellow Fever Susceptibility in New World Nonhuman Primates, Comparison with Humans, and Implications for Surveillance. Emerg Infect Dis. 2021 Jan;27(1):47-56. doi: 10.3201/eid2701.191220. PMID: 33350931; PMCID: PMC7774563.
Mares-Guia, M.A.M.d., Horta, M.A., Romano, A. et al. Yellow fever epizootics in non-human primates, Southeast and Northeast Brazil (2017 and 2018). Parasites Vectors 13, 90 (2020). https://doi.org/10.1186/s13071-020-3966-x
Reply:
Thank you for your suggestions. We cited both articles in this section: “The recent spread of YFV in the entire Southeast and central and eastern part of the South region from 2016 to 2019 has caused thousands of deaths of humans and NHP living close to the most densely populated areas of the Atlantic coast, originating the most severe Brazilian outbreak of the last 80 years [5-11].”
In the third paragraph on page 5, correct to “Sa.” (Sabethes);
Reply:
Text amended as recommended.
at the end of the second paragraph on page 6, are there any details about the epidemiological investigation/confirmation of autochthony of the 4 human cases whose genotypes could not be classified into any of the clades? For the cases of Teresópolis, as mentioned in the discussion, could perhaps be related to the proximity of Maricá and/or high circulation of visitors, but regarding the other two cases is there any information? Well, the dates of the cases are inserted in a very tight sequence with the dates of the surrounding cases. Anyway, intriguing.
Reply:
I agree with the reviewer. It is intriguing. Careful anamnesis was performed for these 4 cases, and no recent travel history was reported. The two cases, that did not group into the R.J. clades (in addition to the two from Teresópolis, already discussed), were in the municipalities of Porciúncula and São Fidélis. Porciuncula directly borders E.S., which could explain its position in this E.S. cluster. São Fidélis is also not very far from the border with ES, and it is not difficult to imagine another viral introduction from ES. Anyway, it is a challenge to infer a possible cause for the fact that the genomes are unclassified.
regarding the results on comparisons between dispersion speeds, in general, for the reader who is not a specialist in statistics (as is my case), the medians seem to satisfactorily describe the differences between each region, especially when we note the range of variation . But something I missed a bit was graphically visualizing the speed variation within the two periods, from February to June 2017 and from October 2017 to April 2018, according to supplementary tables 1 and 2.
Reply:
In our opinion, it would not make much sense to remake these figures in the sample periods (presented in supplementary tables 1 and 2) since the statistics are not based only on the samples but on all ancestral nodes, which are reconstructed in time and space. Thus, the dispersion velocity we calculated considers the geographic coordinates and the date that the program estimated for each ancestral node of the tree, averaging all Bayesian trees (1000 trees). When we plot only the period suggested by the reviewer, there is no considerable variation, as the time interval is small (a few months), and the sample density in this period (considering all locations - RJ, SP, ES, and MG) ends up being low.
in the discussion, in the last paragraph on page 11 the authors referred to “smaller geographic scales”; wouldn’t they be bigger geographical scales? For example 1:50,000>1:500,000. The same can be seen at the beginning of page 13 under “lower scales”.
Reply:
We amended the text to avoid further misunderstanding. We did not mean a literal geographic scale. Instead, we indicated that we are looking at a small portion of a phenomenon that must be bigger than our study. Hence, we changed the word “scale” to “area”.
Reviewer 4 Report
This manuscript about the use of genetic analysis to understand the YFV recent epidemics in southern Brazil is scientifically sound with clear materials and methods and the results are well justified. However, the current manuscript is too long with plenty of duplicated information, it must thus be reduced to about half through eliminating duplicated information. Further, the citation of the name of species does not follow the international standards for citation and all citations must be updated according to the standards. Although the sections on materials and methods, as well as results do not need too much changes, the introduction and discussion need full revision to be more consistent and strait to the point. For some hypothesis it is not clear if they are from published literature or the current findings. Some important consideration such as the role of the roads are almost diluted into different paragraphs and the role of the dispersion of mosquitoes is unproven. In conclusion the introduction and discussion parts needs more work.
Author Response
This manuscript about the use of genetic analysis to understand the YFV recent epidemics in southern Brazil is scientifically sound with clear materials and methods and the results are well justified. However, the current manuscript is too long with plenty of duplicated information, it must thus be reduced to about half through eliminating duplicated information.
Further, the citation of the name of species does not follow the international standards for citation and all citations must be updated according to the standards.
Although the sections on materials and methods, as well as results do not need too much changes, the introduction and discussion need full revision to be more consistent and strait to the point.
For some hypothesis it is not clear if they are from published literature or the current findings. Some important consideration such as the role of the roads are almost diluted into different paragraphs and the role of the dispersion of mosquitoes is unproven.
Reply:
Thanks for the comment. In this new version we have shortened the discussion. Concerning the role of roads in virus dispersion, we now limit it to the discussion on the speeds of viral dispersion and the diversity of clusters found in Teresópolis. The role of mosquito dispersal (induced by several factors such as wind) in the spread of arboviruses including yellow fever is well known (e.g.Causey et al. Am. J. Trop. Med. Hyg. 30 301–312, 1950; Haddow et al.Bull World Health Organ. 1964;31(1):57-69).
In conclusion the introduction and discussion parts needs more work.
Reply: Thank you very much for critically review our manuscript.
We reviewed the manuscript and amended the scientific citations throughout.
Round 2
Reviewer 2 Report
Thanks for your response.